# Quantitative assessment method for firefighting danger based on numerical simulation of forest fire spread in canyon wind fields

**Ao Wang[1], Chenghu Wang**  **[1,2]\*, Guiyun Gao[1,2], Ningyu Wu[1], Haiyan Su[1]**

**1** National Institute of Natural Hazards, Ministry of Emergency Management, Beijing, China, **2** Beijing Engineering Research Center of Earthquake Observation, Beijing, China.

\* huchengwang@163.com

## Abstract

Forest firefighting incidents frequently occur in mountainous and canyon regions which are characterized by complex topography, primarily because of variable local wind patterns that create conditions conducive to the spread of forest fires. This study focuses on the Muli forest fire in Liangshan Prefecture, which occurred on March 28, 2020. The WindNinja modeling software was utilized to simulate the valley wind field, whereas FARSITE modeling software was used to assess the fire spread rate, fireline intensity, and flame length. Moreover, a comprehensive forest firefighting risk assessment model was developed, incorporating factors such as forest fire behavior, fuel types, topography, and vulnerability indices. This model examines the analysis of risk variations across different topographical features and vegetation types. The analytic hierarchy process was employed to determine the weight of each index. Based on the numerical modeling data of forest fire behavior, it was found that the proportion of areas with high and perilous spreading rates in the study area was 16% and 54%, respectively. Areas exhibiting highly and extremely dangerous fireline strengths comprised 5.9% and 0.14%, respectively. Furthermore, the proportion of areas with highly and extremely dangerous flame lengths was 21% and 8.1%, respectively. The average firefighting danger index for grassland was 11, which is the highest among the five forest vegetation types, (evergreen broadleaf forests, evergreen coniferous forests, deciduous broadleaf forests, grasslands, and shrublands), whereas the average danger indices for deciduous broadleaf forest and evergreen coniferous forest were 10.3 and 7.4, respectively. The comprehensive assessment results indicated that 16.5% and 21.4% of the study area face high and extremely high firefighting danger levels, respectively. The comprehensive firefighting danger index for grassland was the highest among all vegetation types in the research area, thereby identifying it as a critical zone for preventing firefighting casualties and implementing countermeasures.

**Data availability statement:** All relevant data are within the paper and its Supporting Information files.

**Funding:** This study was supported by the National Key Research and Development Program of China (Grant No. 2021YFC3001900).

**Competing interests:** The authors have declared that no competing interests exist.

## Introduction

Since the beginning of the 21st century, numerous substantial forest fire events have occurred in the United States, Canada, Australia, and Greece, highlighting the destructive and complex nature of wildfires as natural disasters across various nations [1]. During a forest fire, various factors, including a multifaceted and dynamic wildfire environment, unpredictable fire behavior, abrupt changes in topography, and variable meteorological conditions, can result in deflagrations and other extreme combustion events. Therefore, efforts to combat forest fires involve considerable danger, as even minor lapses in attention can lead to the injury or death of rescue personnel [2]. Wind is a critical factor that influences the fire spread, especially in the alpine and gorge regions, where the local wind patterns are complex and variable owing to the rugged terrain, complicating the predictions of fire spread. Between 2012 and 2022, China experienced 27,091 forest fires, averaging 2,462.8 incidents annually, which resulted in 524 fatalities, or an average of 48 deaths per year [3]. For example, on October 17, 2022, a substantial forest fire in Xintian, Hunan Province, escalated into a deflagration during evacuation efforts, tragically resulting in the deaths of two firefighters [4]. Similarly, on March 13, 2021, a wildfire occurred in Huangshan, Yuanzhou District, Guyuan City, within the Ningxia Hui Autonomous Region. This caused the deaths of two firefighters and injuries to six others owing to a sudden change in wind direction. Furthermore, on March 14, 2019, a forest fire in Qinyuan County, Shanxi Province, trapped seven firefighters because of high wind speeds. Despite extensive rescue efforts, one firefighter sustained minor injuries, whereas the six others tragically lost their lives. Consecutive forest fires in Muli County (March 30, 2019) and Xichang City (March 30, 2020), Sichuan Province, resulted in the deaths of 49 firefighters, leaving a profound and painful lesson for the local communities and society at large [5,6].

Various wildfire spread models and simulation platforms have currently been established in multiple countries and regions [7]. Notable examples include the Rothermel surface fire spread model in the United States, the McArthur Forest Fire Danger Index in Australia, the Fire Behavior Prediction Model in Canada, and Wang's model in China [8,9]. However, the efficacy of fire spread models in real-world scenarios is limited as these models often fail to adequately account for the influence of wind fields on fire behavior and establish a connection between fire dynamics and fluctuating wind conditions. Consequently, a substantial disparity exists between the predictions generated by these models and the actual behaviors of fires. FARSITE integrates the Rothermel surface fire spread model using fundamental physical principles and empirical data. FARSITE uses this principle to break down a fire perimeter into discrete segments ("wavelets") that propagate independently based on local environmental conditions. It computes fire spread rates by considering various factors such as fuel type, meteorological conditions, and terrain slope, achieving a high degree of accuracy through the application of an energy balance equation [10]. In contrast to physical-based models, such as FIRETEC and WFDS, FARSITE is more adept at rapidly simulating large-scale fires while maintaining manageable

scientific and computational costs. Sibanda et al. used the FARSITE and WindNinja models to simulate wildfire behavior in the challenging topography of Nepal, aiming to estimate carbon emissions [11]. Owing to the prolonged simulation times associated with extensive fire areas, Sanjuan et al. proposed a zoning strategy for local wind field calculations, thereby optimizing wind field simulations within time constraints [12]. Seungmin et al. improved the accuracy of FARSITE's fire spread predictions by integrating the directional spread rate adjustment factors and genetic algorithms. These studies have contributed to the development of more effective fire response strategies, which can enhance resource allocation and mitigate the overall impact of fires.

In 2022, the administrative authorities in China have implemented a guiding framework for forest and grassland fire prevention that prioritizes both safety and a people-centered approach. This framework underscores the importance of safeguarding the lives and well-being of the public and firefighters alike [13]. To mitigate the risk of casualties during forest firefighting operations, enhancing targeted prevention and management strategies is imperative. Many experts and scholars are currently leveraging high-resolution remote sensing technology to improve emergency response capabilities related to forest fires, which facilitates the tracking of fire points, monitoring of fire lines, assessment of affected fire areas, and analysis of various disaster-related information from regions impacted by forest fires.[14,15]. Such initiatives empower firefighting teams to effectively manage disaster situations, thereby allowing for critical time to implement effective disaster management strategies. With the increasing use of technology in forest fire spread analyses, numerous countries and regions that are frequently subjected to forest fires have developed various models and simulation platforms. For instance, Qiao et al. created a forest fire spread model using remote sensing and GIS interpretation for the fire analysis; this enhanced the accuracy and reliability of the parameter algorithm and offered remarkable guidance for scientific firefighting and rescue operations.[16]. Reimer et al. analyzed burning probability based on the initial fire intensity, spread rate, and response time [17]. Furthermore, Li et al. established a virtual simulation training system aimed at improving forest firefighting and rescue efforts [18]. The FARSITE model, which has gained remarkable traction for simulating spatially heterogeneous wind fields and intricate terrain configurations. In contrast to static models, FARSITE establishes a dynamic relationship between fire behavior and various environmental parameters, including slope angle, slope aspect, and fuel moisture content, thereby rendering it highly effective for applications in canyons [19].

In the domain of forest firefighting, abrupt alterations in forest fire behavior can be attributed to various factors, including meteorological conditions, topography, and fuel availability. To mitigate the potential loss of personnel and property resulting from these variables, numerous researchers have conducted in-depth investigations into forest fire risk assessments. For instance, Zong et al. formulated a comprehensive method for assessing forest fire risk that incorporated indicators such as the probability of fire ignition and the types of forest fuel present in a given area [20]. Similarly, Yanmei et al. utilized data on wildfire-exposure elements and forested areas, applied area weighting and statistical cluster analysis to categorize forest fire risk across the provinces of China [21]. Although these studies provide valuable insights for understanding forest fire risk management at the macro level, their applicability to practical forest firefighting safety is limited.

Currently, a limited body of research exists dedicated to assessing danger associated with forest firefighting and rescue operations and the intricate characteristics of various fire behavior that influence safety. In addition to certain academic institutions, few organizations and agencies have investigated innovative approaches and technologies pertinent to forest firefighting. Notable exceptions are the Commonwealth Scientific and Industrial Research Organization and the Natural Resources and Forestry Department of Victoria in Australia, which have conducted studies on forest fire behavior [22]. Consequently, the development of quantitative classifications of forest firefighting danger is essential for safeguarding the well-being of firefighters and enhancing the overall efficiency of firefighting efforts. As a decision support mechanism, quantitative danger assessment equips commanders with a basis for making informed strategic and tactical decisions, thereby mitigating the potential escalation of disasters [23].

Considering the unique geographical characteristics and prevalent fire incidents in southwest Sichuan, the aim of this study is to investigate the following research question: In what ways can a coupled WindNinja–FARSITE model, along

with a comprehensive danger assessment model, enhance the quantification of firefighting danger in canyon terrains? To achieve this objective, we propose the following substantial contributions:

(1) Developing a high-resolution wind–fire coupling simulation framework tailored to canyon topography.

(2) Introducing an innovative firefighting danger assessment model that integrates fire behavior, fuel characteristics, terrain features, and exposure indices.

(3) Quantitatively identifying high-danger zones and vegetation types to inform and optimize firefighting strategies.

## Materials and methods

### Overview of the research area

Muli Tibetan Autonomous County is located in the northwestern region of Liangshan Prefecture, Sichuan Province (27°40'–29°10'N, 100°03'–101°40'E), adjacent to the southern portion of Ganzi Prefecture. This area is situated in the intersection of the Yunnan–Guizhou and Qinghai–Tibet Plateaus, characterized by pronounced elevation differences resulting from deep river. The terrain generally slopes from north to south and features prominent mountains and gorges. In 2020, the county recorded an annual average temperature of 22.8°C, an average wind speed of 1.69 m/s, and a total annual precipitation of 1,984.3 mm. The period designated for forest and grassland fire prevention in the county spanned from December 1, 2020, to June 30, 2021. Owing to its unique geographical and climatic conditions, the county has recently experienced notably frequent forest fires, often leading to substantial casualties and property damage.

In the research area, the climate is characterized by pronounced seasonal fluctuations and a high frequency of extreme weather phenomena, especially marked by ongoing drought conditions during the fire season, which typically spans from January to May. The landscape is mainly covered by evergreen coniferous forests, such as Yunnan pine and Pinus yunnanensis, as well as mixed coniferous broadleaf forests, broadleaf forests, shrubs, and tussocks. In addition, interforest meadows featuring remarkable peat bog development are prevalent, increasing the likelihood of underground fires [24]. Owing to its climatic and vegetative attributes, the this area has been designated as a critical national fire risk zone. Recently, several severe forest fire incidents have recently occurred, resulting in substantial casualties and property damage, thereby eliciting considerable societal concern.

On March 28, 2020, at 19:30, a remarkable forest fire was started at the border of Chutouwan Village in Qiaowa Town and Xiangjiao Village in Xiangjiao Township within Muli Tibetan Autonomous County. The area affected by the fire was approximately 230 hectares. On the afternoon of March 29, 2020, a sudden increase in wind speed caused the surface fire to escalate into a crown fire, thereby complicating firefighting efforts. On March 30 and 31, strong winds impeded the effectiveness of the firefighting operations. In the early hours of April 1, firefighting teams capitalized on favorable conditions characterized by low temperatures, high humidity, and minimal wind to combat the blaze. However, in the afternoon of April 1, the northern perimeter of the fire reignited, and under the influence of extreme wind variability, the fire rapidly spread to the northeast and northwest. From April 2–6, firefighting teams undertook a comprehensive operation in the affected area, successfully extinguishing the forest fire by April 7.

### Input data and geoprocessing

Three fundamental data types were required to implement the forest fire spread and WindNinja models: digital elevation model (DEM) data, land cover data, and meteorological data. The DEM data were sourced from the Geographic Spatial Data Cloud (http://www.gscloud.cn) and then processed using spatial analysis techniques to yield slope and aspect data with a 30-meter spatial resolution. The land cover data were obtained from the global 30-meter spatial resolution fine land cover product (GLC_FCS30–2015) published by Liu Liangyun's research team in 2019 [25]. The meteorological data were collected from the European Centre for Medium-Range Weather Forecasts, using the recorded meteorological data from

the nearest weather station to the Muli fire site (station number 56462; 29°N, 101.5°E) to inform the surface average wind field. This study also incorporated satellite imagery from Landsat 8 (http://www.gscloud.cn/) to accurately delineate areas affected by Muli fire. Before integrating this data into the model, it underwent a series of preprocessing steps, including radiometric correction, image registration, image cropping, and image mosaicking, to ensure uniform spatial extent across all datasets.

### Research methods

**Wildfire simulations.** Wildfire propagation simulations were performed utilizing the coupled WindNinja–FARSITE framework. FARSITE [19] is a two-dimensional fire modeling system that serves as a valuable supplementary decision-making instrument for the prevention and management of forest fires [10]. WindNinja was used to produce high-resolution wind fields by incorporating the DEMs and meteorological data [26]. A significant benefit of WindNinja is its ability to integrate seamlessly with FARSITE, allowing for the direct application of the simulated wind fields [27].

**Comparison and selection of the forest fire behavior index.** To analyze the fire behavior indicators associated with the March 28 Muli forest fire, ArcGIS was used to generate landscape files that incorporated elevation, slope angle, slope aspect, and forest fuel characteristic data within a consistent resolution and spatial extent [24]. Using remote sensing imagery and prior research findings, the forest fuel types within the study area were categorized as follows: evergreen broadleaf forests, evergreen coniferous forests, deciduous broadleaf forests, shrubs, and grasslands. The assignment of fuel types was conducted based on the 40-fuel model system established by Scott [28], including specific models such as M186 for high-load broadleaf litter. The fuel type model assignments were validated using remote sensing image.

Wildfire behavior encompass the spatial and temporal dynamics that occur during the fire-spreading process across different fuel complexes, ranging from the initial ignition phase to the point of extinction. Wildfire behavior is characterized by several factors, including the fire rate of spread, the expansion of the fire perimeter, the escalation of the fireline intensity, and extreme fire behavior such as spotting, fire whirls, and flare-ups [29]. Herein, the following specific indices were used as metrics for the danger assessment:

(1) The spread rate of a wildfire, recognized as a fundamental indicator of fire behavior, significantly impacts the development and expansion of the fire. Furthermore, it is an essential metric for evaluating the dynamic alterations that occur within the fire environment during firefighting efforts. This spread rate also serves as a primary criterion for fire command personnel when organizing firefighting teams [30].

(2) Fireline intensity refers to the amount of heat energy released per unit of time when the flame propagates 1 meter from the surface of the area that is burning to the leading edge of the fire. The intensity is influenced by the spread rate of the fire and the heat released per unit area [31].

(3) Flame length is defined as the distance from the base of the flame at the center of the forest fuel center to the average tip of the flame, a measurement that is influenced by the angle formed between the flame and the ground surface. This parameter is closely associated with the intensity of the fire line and serves as a vital metric for assessing firefighting efforts [31].

**Firefighting danger assessment model.** The quantitative assessment of fire danger associated with forest firefighting substantial complexity. Through expert consultation, four primary danger assessment indices were identified: forest fire behavior, fuel factor, terrain factor, and exposure (Table 1), which collectively address the fire dynamics and contextual vulnerabilities (Fig 1) [32]. The proposed model incorporates three forest fire behavior as secondary evaluation indices (spread rate, fire line intensity, and flame length), two fuel factors as secondary indices (fuel type and fuel density), two terrain factors as secondary indices (slope angle and slope aspect), and two exposure types as secondary indices (distance to town and distance to the water source) (Table 1).

**Table 1. Classification of each secondary index and danger index.**

| Forest fire behavior | Fuel factor | Score | Rescue danger level | Terrain factor | Exposure | Score | Rescue danger level |
|---|---|---|---|---|---|---|---|
| Spread rate Fireline intensity Flame length | Fuel type Fuel density | 0 | None | Slope angle Slope aspect | Distance to town Distance to the water source | 0 | None |
| | | 5 | Low | | | 2 | Low |
| | | 10 | Medium | | | 6 | Medium |
| | | 15 | High | | | 8 | High |
| | | 20 | Extreme | | | 10 | Extreme |

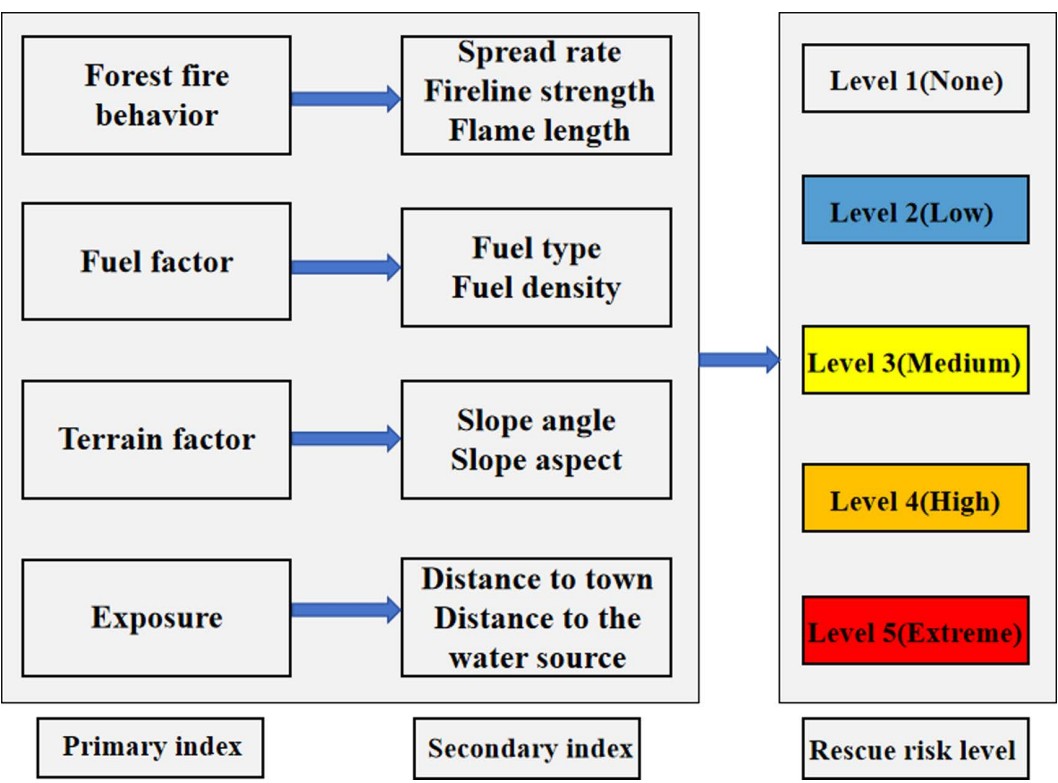

**Fig 1. Assessment model of forest firefighting danger level.**

The proposed model uses the analytic hierarchy process (AHP) to determine the weights of the assessment indices and develop a robust model for evaluating forest firefighting danger as follows [33].

$$X_i = \sum b_j Q_j \tag{1}$$

where $X_i$ is obtained by the weighted calculation of the risk index of the secondary index, $Q_j$ is obtained from Table 1, and $b_j$ is the weight coefficient of the secondary index.

$$Y = \sum a_i X_i \tag{2}$$

where $Y$ is the comprehensive index of the forest firefighting risk, $X_i$ is the value of each level index, and $a_i$ is the weight coefficient of the first-level index.

A comprehensive index scoring methodology was used to establish the firefighting danger index, as detailed in Table 1. The natural breakpoint method was utilized to categorize the fire danger into five distinct levels: none, low, medium, high, and extreme danger. Scores were allocated based on the thresholds derived from historical firefighting incident data and expert assessments, with reference to "Forest and Grassland Firefighting Safety Studies" by Shu et al. For instance, flame lengths exceeding 3 meters (indicative of extreme danger) were assigned the highest score owing to their association with firefighter fatalities. As the behaviors of forest fires and associated fire-weather conditions deteriorate, the likelihood of extreme fire events escalates, thereby complicating firefighting efforts and heightening the danger index. This study used fuel type and fractional vegetation coverage(FVC) as primary indicators for assessing firefighting danger. Based on the prevalent vegetation in the study area, the fuel types were classified into evergreen broadleaf forests, evergreen coniferous forests, deciduous broadleaf forests, grasslands, and shrublands.

Exposure, primarily assessed by evaluating the proximity of fire to towns and water sources, serves as the basis for ranking the level of firefighting danger. A positive correlation exists between these variables; in particular, a shorter distance corresponds to a reduced firefighting danger, thereby yielding a lower danger score.[20].

In AHP, the weight of each index is determined by the eigenvector associated with the maximum eigenvalue of the judgment matrix, as outlined in reference [32]. The AHP weights were obtained through expert surveys that assessed pairwise comparisons of the indices. The consistency ratios, verified by Saaty's methodology, were assessed as less than 0.1 [34].The weights assigned to the indicators within each layer are detailed in Table 2.

## Results

### Modeling of canyon wind field simulation

To analyze the characteristics of the canyon wind field prior to and following the forest fire, the wind speed and direction data from March 25 to April 3 were used for statistical examination (Fig 2). The findings indicated a distinct daily periodicity, characterized by a gradual increase in wind speed from early morning to afternoon, typically culminating in a peak between 2 and 4 PM. Transitional before morning and evening were associated with a slight decrease in wind speed. Importantly, a remarkable increase in wind speed was observed during the Muli fire, which contributed to conditions conducive to the fire's spread. In addition, the wind direction before the fire predominantly was NWW (281°–348°) during the day and (348°–78°) at night. By contrast, postfire wind direction exhibited considerable variability, primarily oscillating

**Table 2. Weights of forest firefighting danger assessment index.**

| Primary index | Primary index weight |
|---|---|
| Forest fire behavior | 0.28 |
| Fuel factor | 0.48 |
| Terrain factor | 0.17 |
| Exposure | 0.06 |
| **Secondary index** | **Secondary index weight** |
| Spread rate | 0.61 |
| Fireline intensity | 0.30 |
| Flame length | 0.09 |
| Fuel type | 0.83 |
| Fuel density | 0.17 |
| Slope angle | 0.83 |
| Slope aspect | 0.17 |
| Distance to town | 0.25 |
| Distance to the water source | 0.75 |

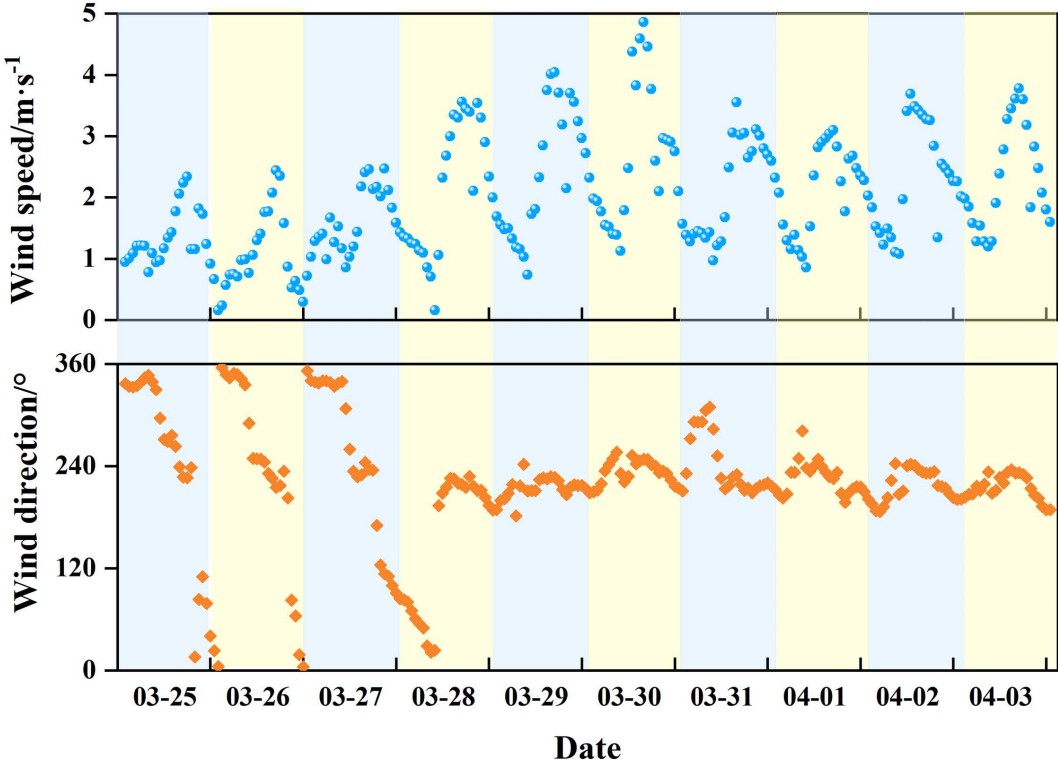

**Fig 2. Wind field data from March 25 to April 3, 2020 in the research area.**

between the SSE and SWW directions (168°–258°). Throughout the analyzed period, the prevailing weather conditions were predominantly clear, characterized by minimal cloud cover and low levels of precipitation.

In the context of this research, data obtained Meteorological Observatory No. 56462 (29°N, 101.5°E), which is situated in proximity to the Muli fire site, was used for comparison with the hourly wind field data generated by the WindNinja model during the period of fire spread (Fig 3). The comparative analysis of the simulated and observed values from the WindNinja model and the weather station is illustrated in Fig 3. The wind speed predicted by the WindNinja model was approximately 2 m/s greater than the actual observed measurements, suggesting that the canyon topography contributed to an increase in wind speed owing to the narrow tube effect. The trends in variation for both datasets were consistent. with elevated wind speeds occurring in the afternoon and a stabilization of the wind field during nighttime. Furthermore, the simulated wind direction closely aligned with the observed wind direction, predominantly remaining stable within the range of 210°–250°, with SW identified as the prevailing wind direction. Overall, the simulation outcomes from the Wind-Ninja model demonstrate the capability to accurately represent canyon wind fields and effectively reflect the dynamic trends of actual wind conditions.

To analyze the evolution of the canyon wind field during the progression of the forest fire, simulations were conducted at six critical time points: 19:00 and 13:00 on March 28, 29, and 30, as well as 19:00 on March 30 and 13:00 on March 31. These time points strategically encompassed the phases from the onset to the spread of the fire, thereby facilitating the examination of wind field characteristics in the study area during diurnal and nocturnal periods. Concurrently, the intensity of the fire line was simulated at each designated time, in conjunction with the wind field analysis. The spread of the forest was predominantly influenced by the southwest wind, with remarkable increases in fire line intensity observed in regions characterized by elevated wind speeds. At 19:00 on March 28, the Muli wind field exhibited a relatively chaotic scattering

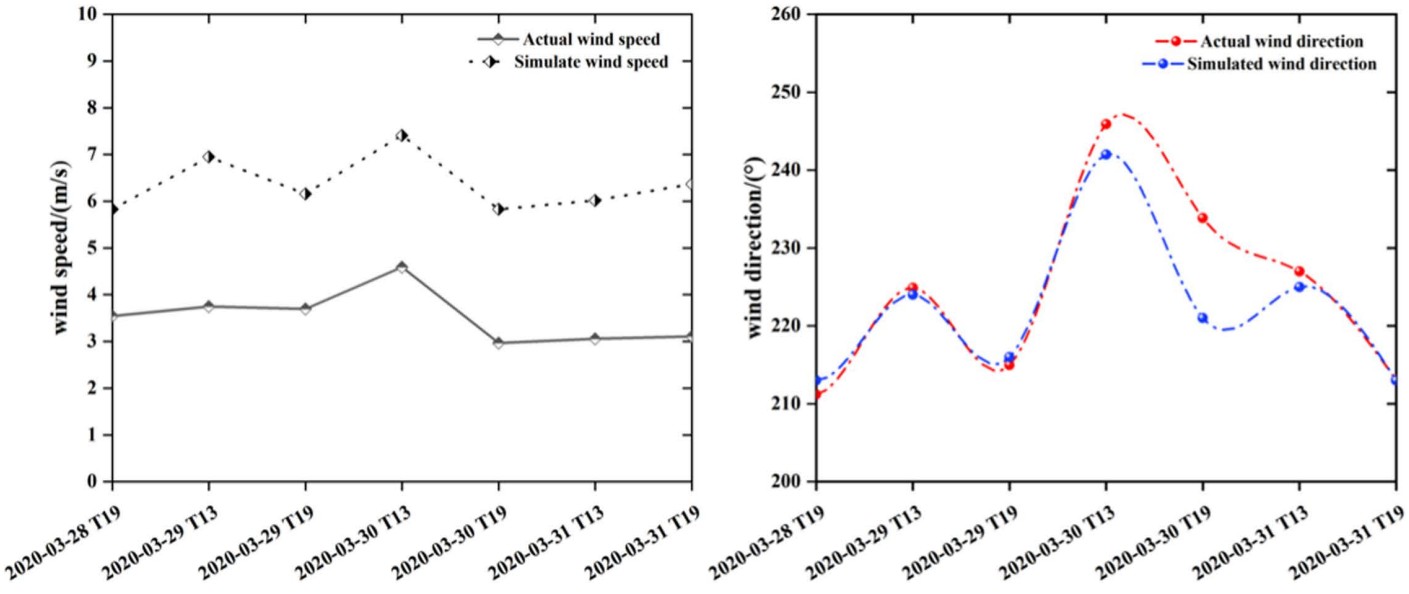

**Fig 3. Comparison between simulated wind field and actual wind field.**

pattern, which provided foundational conditions for the initial spread of the fire. By 13:00 on March 29, the Muli wind field displayed a southerly wind trend influenced by the uphill winds originating from the Xiaojinhe River Valley to the south. The wind speed experienced a marked increase, peaking at 18m/s, which facilitated the northward spread of the forest fire to the north. At 13:00 on March 30, the wind speeds continued to rise, with the southerly winds persisting over the fire site. However, by 19:00 on March 30, both the overall wind speed and fire line intensity remarkably diminished, indicating a potentially optimal window for extinguishing the forest fire. By contrast, at 13:00 on March 31, wind speeds escalated once more, enabling the forest fire to extend northeastward and into broader areas.(Fig 4)

In conclusion, the WindNinja model effectively captures the spatio-temporal variability of the local wind field in canyon topography and exhibits a robust coupling simulation effect with associated fire behavior.

## Coupled forest fire spread simulation

High-precision simulated wind field data were generated using the WindNinja model. These data were subsequently incorporated into FARSITE to assess the impact of spatially variable wind field data on fire propagation. In a comparative experiment, the average surface wind field was employed as the input for the forest fire spread model, while maintaining other parameters constant. During the initial phase of the fire, the simulated mean surface wind field extended beyond the actual fire field boundary in the northern and southeastern regions. The simulation outcomes improved following the integration of simulated wind-field data from WindNinja. However, in the later stages of the fire, both wind field datasets were inadequate in accurately predicting the spread in the northeastern region. In general, the simulation results of the forest fire using the WindNinja wind field were closely corresponded with the actual fire field in terms of the extent of spread, thereby enhancing the overall accuracy of the simulation (Table 3).

## Evaluation indicator analysis

In the study area, the average rates of speed, fire line intensity, and flame length were recorded at 7.2 m/min, 623 kW/m, and 1.4 m, respectively. Regions classified as having high and extremely dangerous danger levels for spread rate

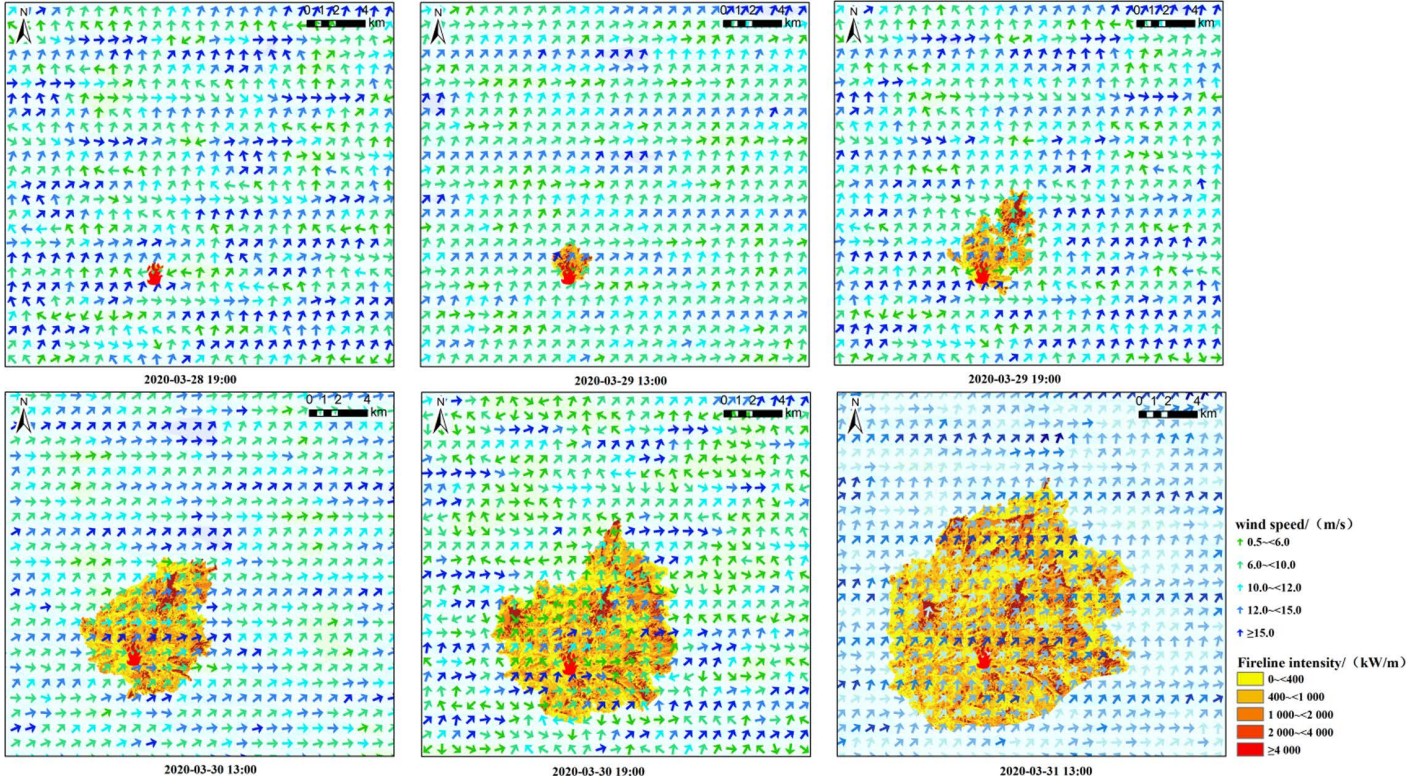

**Fig 4. WindNinja wind field and fire line intensity coupling simulation.**

**Table 3. Comparison of simulation accuracy of different wind fields.**

| Wind field | Time | Area of coincidence area/km² | Over-simulated area/km² | Un-simulated area/km² | SC coefficient |
|---|---|---|---|---|---|
| Average surface wind field | March 29, 19:00 | 60 | 19 | 6.1 | 0.82 |
| | March 30, 19:00 | 86 | 59 | 10.8 | 0.71 |
| | April 1, 19:00 | 78 | 204 | 22 | 0.40 |
| WindNinja wind field | March 29, 19:00 | 62 | 4.4 | 3.9 | 0.94 |
| | March 30, 19:00 | 86 | 36 | 12 | 0.78 |
| | April 1, 19:00 | 80 | 178 | 21 | 0.44 |

comprised 16% and 54% of the total area, respectively, with a predominant concentration in the central southern and eastern sections of the study area(Fig 5a). High-danger areas for fire line intensity represented 5.9% of the total, whereas those categorized as extreme danger constituted 0.14%, exhibiting a more dispersed distribution(Fig 5b). The areas identified as highly dangerous concerning flame length accounted for 21% and 8.1%, respectively, displaying a spatial distribution that closely mirrors that of the fire spread rate(Fig 5c).

The predominant combustible vegetation types within the fire-affected regions were primarily evergreen coniferous forests, grasslands, and deciduous broadleaf forests, which comprised 52%, 46%, and 2% of the total fire area, respectively. The average fire behavior index for grasslands was recorded at 11, representing the highest value among the five categories of forest fuels. This was followed by deciduous broadleaf forests and evergreen coniferous forests, which exhibited average fire behavior indices of 10.3 and 7.4, respectively.

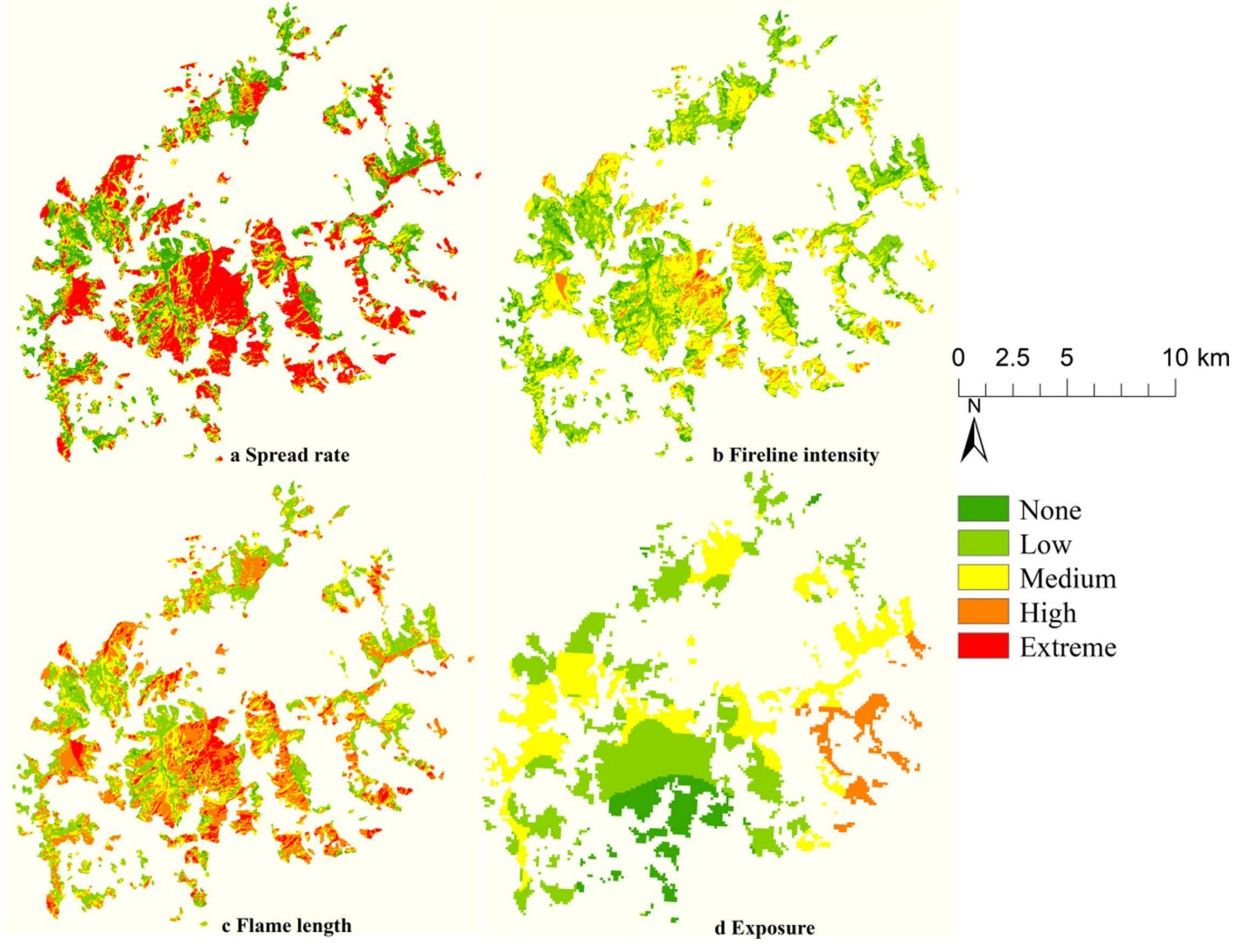

**Fig 5. Simulation results of fire behavior and exposure in the research area.**

The mean exposure index within the study area is recorded at 5.7. The evergreen coniferous forest exhibited the highest exposure index at 5.6, followed by grassland with an index of 5.4, whereas the broadleaf deciduous forest presented the lowest exposure index at 5.3. The distribution of danger levels is categorized as follows: areas with no danger accounted for 11.6%, low danger for 15%, medium danger for 51%, high danger for 16%, and extreme danger for 6.5%. Furthermore, regions characterized by low exposure indices were predominantly located near the Xiaojin River and certain villages in the southern segment of the research area. By contrast, the exposure levels tended to increase with distance from urban centers or water bodies (Fig 5d).

### Comprehensive assessment of firefighting danger

The comprehensive assessment of forest firefighting danger levels is derived from an analysis of various factors, including forest fire behavior, fuel characteristics, topography, and exposure indices, along with the relative weight assigned to

each index. In the context of the Muli forest fire that occurred on March 28 in Sichuan, regions classified as having high and extremely high-danger levels constituted 16% and 21% of the total area, respectively. These high-danger zones were primarily located in the central and western sections of the study area, with a minor presence in the eastern and northeastern regions. Conversely, the majority of the southern and northeastern areas exhibited low to medium danger levels, representing 50% of the total area. In addition, regions classified as having no danger levels accounted for 12%, displaying a scattered distribution(Fig 6).

According to the forest fuel classification, the overall forest firefighting danger indices for grassland, evergreen coniferous forest, and deciduous broadleaf forest were determined to be 10.1, 6.0, and 8.4, respectively (Table 4). Grassland exhibits the highest level of danger, as its average fire behavior index surpasses that of other vegetation types. Consequently, grassland represents a critical focus for monitoring, early warning systems, and danger strategies.

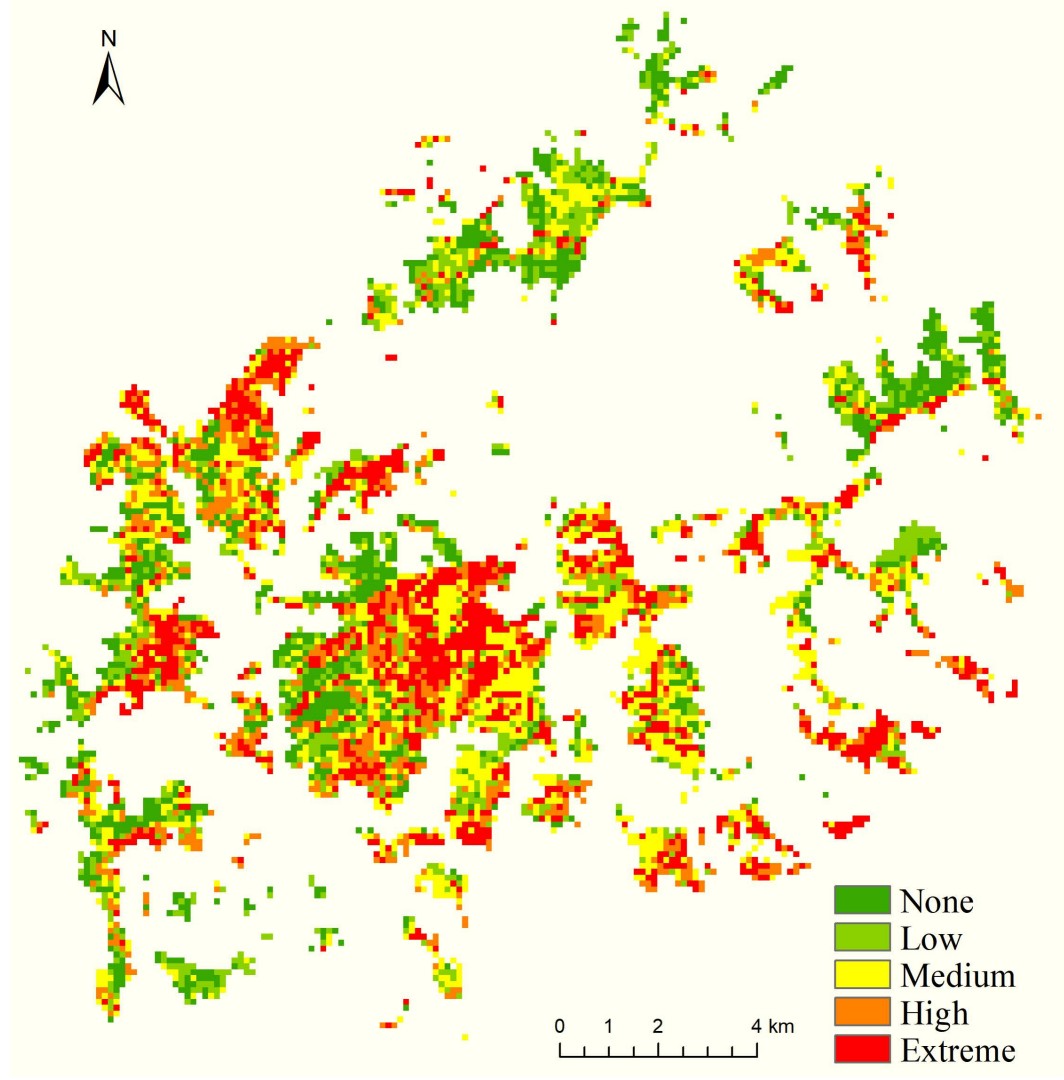

**Fig 6. Forest fire comprehensive danger level map.**

**Table 4. Comparison of simulation accuracy of different wind fields.**

| Vegetation type | Average fire behavior index | Exposure | Comprehensive rescue danger index |
|---|---|---|---|
| Evergreen coniferous forest | 7.4 | 5.6 | 6.0 |
| Deciduous broadleaf forest | 10.3 | 5.3 | 8.4 |
| meadow | 11 | 5.4 | 10.1 |

## Discussion

The precise forecasting of wildfire propagation and behaviors is essential for effectively managing wildfire containment efforts. This study assessed the efficacy of the coupled WindNinja–FARSITE model for simulating the March 28, 2020, Muli forest fire in Sichuan Province. The model accurately replicated the fire spread direction and fire perimeter morphology within the designated study area. However, notable discrepancies were identified between the simulated and actual fire perimeters, which may be attributed to potential inaccuracies in parameterizing the fuel factors, wind field inputs, and the meteorological data used. The intricate topography of the region plays a critical role in shaping local wind patterns, with simulations of canyon wind fields demonstrating greater accuracy than those based on surface-averaged wind fields [10]. Integrating the wind data generated by WindNinja into the FARSITE model enhanced the consistency of the simulations and minimized relative errors. These results highlight the critical need for high-resolution canyon wind field data to ensure the reliable prediction of fire spread. However, the WindNinja model exhibited limitations for simulating wind fields on leeward slopes, likely owing to simplifications inherent in its algorithms. Future research will investigate alternative wind modeling approaches, such as WindWizard [35] and WindStation [36], to improve the spatial resolution of wind vector calculations.

The meteorological data used in this study were obtained from weather stations proximal to the fire site. The spatial and elevational differences between the locations of these stations and the fire zone may have introduced potential biases. Un-simulated areas may also exhibit discrepancies between the input meteorological conditions and local micro-climates. In addition, the areas of over-predicted fire spread may have resulted from unaccounted suppression activities that limited the progression of the actual fire [27]. To enhance the accuracy of future studies, prioritizing the calibration of fuel types, terrain factors, and meteorological drivers is essential. Conducting repeated simulations across various regions in southwestern China while using comprehensive historical fire data will further validate the precision and applicability of the coupled WindNinja–FARSITE model.

Muli County, located within the high-altitude canyon systems of western Sichuan, faces remarkable firefighting challenges owing to wind variability influenced by terrain and the unpredictable behaviors of fire. The catastrophic the 2019 Muli County and 2020 Xichang City wildfires underscore the pressing need for accurately modeling fire spread in complex topographical settings to guide management and rescue efforts. Importantly, the fuel models used by FARSITE were originally designed for North American ecosystems, which exhibit distinct climatic and ecological characteristics unlike those of the subtropical environment of southwestern China. Previous research indicated that implementing region-specific fuel models can substantially improve the efficacy of FARSITE [37]. Therefore, developing customized fuel models specifically adapted to the subtropical forests of southwestern China is crucial for enhancing the accuracy of fire spread simulations in future studies.

Forest fire management, recognized as a crucial global issue, is influenced by various factors, including fuel type, meteorological conditions, and topography. The advancement of scientific forest fire suppression knowledge is contingent upon accumulating practical experience. Historically, the assessments of forest fire risk have primarily relied on weather data and the temporal and spatial distribution of fire incidents [38]. Although pre-disaster factors that contribute to the occurrence of forest fire occurrences have been acknowledged, there is a lack of quantitative analysis of the

phenomena and characteristics associated with forest fire behavior during a disaster, such as spread rate, intensity, and flame length. Currently, most assessments are based on empirical observations. Moreover, in a given region, the forest fire risk tends to remain relatively constant, whereas the danger associated with fire suppression can considerably fluctuate. Although models that simulate fire behavior have demonstrated effective applications in danger assessment simulations, recognizing the inherent uncertainties associated with such simulations is important. The fire suppression process is influenced by numerous factors, including the overall competency of the firefighting personnel and the availability of on-site technical equipment, which are not comprehensively integrated into current assessments. To enhance the objectivity and scientific rigor of these evaluations, these factors could be incorporated into models as correction variables. The forecasting of forest fire suppression danger involves predictions based on objective factors and aims to identify trends related to forest fire incidents through specific mathematical methodologies [22]. This serves as a crucial technical reference for evaluating the danger associated with forest and grassland firefighting. Such predictions play a vital role in guiding firefighting personnel by assessing and identifying the danger levels of forest fires on the day of the incident and over subsequent days.

## Conclusion

To address the limitations of conventional forest fire spread models for predicting fire behavior in complex terrains such as mountainous and valley regions, this study introduces a coupled simulation approach that integrates the FARSITE forest fire spread simulation system with the high-resolution wind field model WindNinja. Furthermore, a forest firefighting danger assessment model was developed, incorporating factors such as forest fire behavior, fuel characteristics, topographical features, and exposure indices. This model was used to quantify the firefighting danger associated with the Muli forest fire that occurred on March 28 in Liangshan, Sichuan Province. The key findings, significantly enriched by the detailed simulation data, offer a robust scientific foundation for implementing targeted forest firefighting strategies and address the core research question on enhancing danger quantification in canyon terrains:

(1) The wind field within the canyon displays a significant diurnal variation, with maximum wind speeds generally recorded between 2 PM and 4 PM. A comparative assessment of the simulated wind speeds against actual local meteorological conditions revealed inconsistencies, with an average deviation of approximately 2 m/s in the simulated data. However, the wind direction exhibits a degree of stability, predominantly blowing from the southwest (SW) and west-southwest (WSW). The high-resolution wind field data produced by the WindNinja model demonstrates its efficacy in accurately simulating the dynamics of forest fire propagation in relation to the canyon's topographical characteristics.

(2) The simulation indicated concerning levels of fire behavior risk, with regions categorized as having high and extremely dangerous fire spread rates accounting for 16% and 54% of the study area, respectively, primarily located in the central and southern regions. Although areas exhibiting high-danger fire line intensity were notable at 5.9%, extreme-danger intensity zones, while less prevalent at 0.14%, pose significant flashpoint hazards. Additionally, high and extreme-danger flame lengths encompassed 21% and 8.1% of the area, respectively, demonstrating a strong spatial correlation with high spread rates. This quantitative spatial mapping of zones exhibiting extreme fire behavior represents a significant finding of the integrated model.

(3) The analysis of fuel types has identified grasslands, deciduous broadleaf forests, and evergreen coniferous forests as the primary combustible materials. Notably, grasslands demonstrated the highest average fire behavior index (11) and an overall firefighting danger index (10.1), followed by deciduous broadleaf forests with indices of 10.3 and 8.4, respectively. These values significantly surpass the danger levels associated with evergreen coniferous forests, which recorded indices of 7.4 and 6.0. This finding highlights the considerable risk posed by grassland fuels in the region. Additionally, exposure analysis revealed that the threat of firefighting danger escalates with increasing distance from urban areas or water sources.

(4) Upon synthesizing all relevant factors, the thorough danger assessment indicated that a significant 37% of the Muli fire site area is subjected to high or extremely high levels of firefighting danger, with concentrations predominantly located in the central and western regions. An analysis based on forest fuel types clearly identified grassland ecosystems as the vegetation type with the highest associated risk.

At the Muli fire site, 38% of the area is classified as having a high or extremely high danger of fire, with this danger predominantly located in the central and western regions of the site. An examination of forest fuel classifications reveals that grassland ecosystems represent the highest-risk vegetation type, exhibiting a comprehensive danger index of 10.1. Areas classified as extreme danger are predominantly located in central and western canyons, where southwesterly winds exceeding 18 m/s have been shown to increase fire spread rates by 54%. To reduce the risk of firefighter casualties, we propose the following recommendations: (1) The real-time integration of WindNinja-FARSITE forecasts into firefighting operations. (2) The implementation of preemptive fuel reduction strategies in grasslands situated within a 5 km radius of urban areas. (3) The tactical avoidance of leeward slopes during peak diurnal wind periods, specifically between 14:00 and 16:00.

## Supporting information

**S1 Fig. Fuel classification.**
(TIF)

**S2 Fig. Research area DEM.**
(TIF)

**S1 Data. Meteorological data.**
(XLSX)

**S1 File. Supporting information.**
(DOCX)

## Acknowledgments

The authors would like to thank the editor and anonymous reviewers for their comments and suggestions, which helped a lot in making this paper better.

## Author contributions

**Funding acquisition:** Chenghu Wang, Guiyun Gao.

**Investigation:** Ningyu Wu, Haiyan Su.

**Project administration:** Chenghu Wang, Guiyun Gao.

**Writing – original draft:** Ao Wang.

**Writing – review & editing:** Ao Wang, Chenghu Wang.

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
