## [Decision Letter · Decision Letter 0]

Dear Dr. Wang,

Thank you for submitting your manuscript to PLOS ONE. After careful consideration, we feel that it has merit but does not fully meet PLOS ONE’s publication criteria as it currently stands. Therefore, we invite you to submit a revised version of the manuscript that addresses the points raised during the review process.

We look forward to receiving your revised manuscript.

Kind regards,

Isidoro Russo, Ph.D.

Academic Editor

PLOS ONE

Journal requirements: When submitting your revision, we need you to address these additional requirements. 1. Please ensure that your manuscript meets PLOS ONE's style requirements, including those for file naming. The PLOS ONE style templates can be found at https://journals.plos.org/plosone/s/file?id=wjVg/PLOSOne_formatting_sample_main_body.pdf and https://journals.plos.org/plosone/s/file?id=ba62/PLOSOne_formatting_sample_title_authors_affiliations.pdf. 2. Please note that PLOS ONE has specific guidelines on code sharing for submissions in which author-generated code underpins the findings in the manuscript. In these cases, we expect all author-generated code to be made available without restrictions upon publication of the work. Please review our guidelines at https://journals.plos.org/plosone/s/materials-and-software-sharing#loc-sharing-code and ensure that your code is shared in a way that follows best practice and facilitates reproducibility and reuse. 3. We note that the grant information you provided in the ‘Funding Information’ and ‘Financial Disclosure’ sections do not match.  When you resubmit, please ensure that you provide the correct grant numbers for the awards you received for your study in the ‘Funding Information’ section. 4. Thank you for stating the following financial disclosure:  [National Key Technologies Research & Development Program].  Please state what role the funders took in the study.  If the funders had no role, please state: ""The funders had no role in study design, data collection and analysis, decision to publish, or preparation of the manuscript."" If this statement is not correct you must amend it as needed. Please include this amended Role of Funder statement in your cover letter; we will change the online submission form on your behalf. 5. Please include a copy of Table 4 which you refer to in your text on page 12. 6. Please include captions for your Supporting Information files at the end of your manuscript, and update any in-text citations to match accordingly. Please see our Supporting Information guidelines for more information: http://journals.plos.org/plosone/s/supporting-information. 

Reviewers' comments:

Reviewer's Responses to Questions

**Comments to the Author**

1. Is the manuscript technically sound, and do the data support the conclusions?

Reviewer #1: No

Reviewer #2: Yes

2. Has the statistical analysis been performed appropriately and rigorously?

Reviewer #1: No

Reviewer #2: Yes

3. Have the authors made all data underlying the findings in their manuscript fully available?

Reviewer #1: No

Reviewer #2: Yes

4. Is the manuscript presented in an intelligible fashion and written in standard English?

Reviewer #1: No

Reviewer #2: Yes

Reviewer #1: Manuscript title: Quantitative assessment method for firefighting risk based on numerical simulation of forest fire spread in canyon wind fields

Manuscript number: PONE-D-24-40817

General comments:

In the Methods section, the description of the input data is clearly insufficient. The conceptual risk assessment model does not make sense. Authors should start by conceptually defining the risk model based on the state of the art. This conceptual framework is critical. In order to be able to transform a generic "fire risk" model into a "firefighting risk" model, you must define the indicator classes based on suppression capacity. There are many inconsistencies and even errors in the use of fire science terminology. There are factors that enter the model twice. The way you defined the classes is wrong, it is disconnected from the physical process of fire propagation. It is also unclear how the factor weights were established. You simulated moments of a single fire (why this fire?). You did not describe the fire regime in your study area, nor the fire you used as a reference. You did not include the parameters used in the simulation, nor did you compare the simulated fire with the reference fire. This fire should have only been used to calibrate the parameters required by the simulator. The results should be evaluated using metrics such as overall accuracy, true skill statistics or others (see the work of Pontius on accuracy metrics). Then, these parameters would be used to define simulation scenarios. For each scenario, N fires would be simulated and the AHP would receive fire behavior metrics as input data, as well as the infrastructure frequency reached by the simulated fires.

You have a "Results and Discussion" section and a "Discussion" section, but the current discussion has not been done. To what extent does your proposal differ from others? Figures 8 and 9 are not understandable (why was the assessment not carried out for the entire study area?), and once again you are not consistent in terminology (what is "firewire strength"?). In the end, you have little more than a simulation that it is not even possible to know if it was done well.

The manuscript is very immature, and is not yet of sufficient quality to be considered for publication.

Detailed comments:

L108: What do you mean by “climate is complex”?

L109: By "fire protection period" do you mean "fire season"? If so, what period covers the "fire season" in your geographic context?

L110: Please change “The vegetation types primarily comprise (…)” by “The landscape is mainly covered by (…)”

L111: What do you meany by “serious swamping”? Please clarify. Will there be more peat bogs than swamps?

L112: Please change “underground fire occurrence” by “underground fires”

L115: Change to “Input data and geoprocessing”

L116-120: The description of the input data has to be more detailed. For example, it is important to mention what input data allowed generating the DEM, what methods were used to generate the derivatives that give rise to the slope and aspect (Horn's method?) and what software was used. The designation "land type" is unspecific, apparently it is a land cover map whose metadata must be minimally described (number of classes, method used, input data and auxiliary data that supported the construction of the dataset). Are meteorological data reanalysis? What scale are they on?

L122: Change the title to “Wildire simulations”

L123: Add a reference of Mark Finney to FARSITE

L124: Add "fire" before "spread"

L124-126: Please rewrite to improve readability

L123: Add a reference to WindNinja

L127-129: "In addition" why? Simulations can be carried out with surface wind simulation or not. This choice is part of the simulation process, not exactly something independent.

L129-130: Flammap already does this integration without having to import WindNinja outputs

L131-138: Move to the previous section and rewrite to improve readability

L132: Change “slope” by “slope angle” and “slope direction” by “slope aspect”

L135: What are the "classic fuel model values"? NFFL? Scott & Burgan? The reclassification of land cover to fuel models needs to be better described and should be supported by photographs.

L142: Change “Forest fire behavior encompasses the various changes that occur throughout the fire-spreading process of forest fuel (…)” by “Wildfire behavior encompasses the spatial and temporal dynamics that occur throughout the fire-spreading process across different fuel complexes (…)”

L143-144: Please be careful in your use of terminology:

"spreading speed of the fire head" = "fire rate of spread"

"fire intensity" = "fireline intensity"

"firefield scope expansion" = ?

"other extreme fire behaviors" = ? (probability of crown fire?)

L147: The flame length is directly proportional to the fireline intensity, but not to the fire rate of spread.

L154-156: Move to the next section

L158-159: You state that “fire behavior, forest fuel factor, topography, and exposure were identified as the four first-level indexes”. How? What other factors did you consider? Topography and fuels are already integrated into fire behavior indicators. You are "doubling" the weight of these factors in the analytical process.

L160: You have to maintain consistency in the terminology used

L161: What do you mean by “secondary evaluation indexes”?

L161: “Fuel type” or fuel models?

L161: “Fuel type and intensity”? What’s “fuel intensity”?

L162: "Slope" and "slope inclination" are basically the same. Maybe you mean "slope angle" and "slope aspect"

L163-164: The distance to a burnt edge is not exactly a measure of exposure. Infrastructure exposure can be assessed by the frequency of intersections by simulated fires. See the works of Bruno Aparício and Alan Ager.

L171: The scores in Table 1 are anything but comprehensive.

L171-172: The use of natural breaks is not recommended to define risk classes. Class thresholds must be defined according to the ecological process under evaluation. You have resorted to a purely statistical form of division without any connection to the physical process of fire spread or the fire suppression capacity.

L172: “Fire risk” and “fire danger” are not the same thing. See the seminal works of Hardy, Bachmann or Keane

L173: Change “meteorological” by “fire-weather”

L174-175: But you do not present any of the fuel structure indicators that you mention

L176: How have you measured “fuel density”?

L176-178: Repeated

L184-187: The use of the AHP method is interesting and appropriate, but you do not define how the factor weights were established

Reviewer #2: This manuscript constructs a forest firefighting risk assessment model to quantify comprehensive risk based on forest fire behavior, forest fuel, topography, and exposure indexes. The authors also demonstrate the potential of using FARSITE and WindNinja to provide fire behavior information for their risk assessment model.

Here are some considerations:

Line 52-63: The authors adopted FARSITE as the fire spread model in this study, yet it was not mentioned in the literature review section. I suggest enhancing the literature review to further discuss various fire spread models.

Line 91-97: The research question and unique contribution are not explicitly stated. I recommend revising this paragraph to clearly articulate the research question and contributions.

Line 116-120: The topographic and fuel data were extracted from different sources. It would be helpful to mention if the resolution is consistent or if re-sampling was used. What is the resolution used in this study?

Line 131: More information could be provided for the simulated March 28 Muli Forest fire, such as crown fire activity, burning periods, simulation duration, and fuel moisture.

Line 135: Assuming the authors classified fuels based on the 40 Scott and Burgan model, it would be helpful to mention this and cite accordingly.

Line 171-173: Further elaboration of the score assignment is needed. How were the breakpoints decided, and how were those characteristics transferred to scores?

**Do you want your identity to be public for this peer review?** For information about this choice, including consent withdrawal, please see our Privacy Policy

Reviewer #1: **Yes: ** Nuno Ricardo Gracinhas Nunes Guiomar

Reviewer #2: No

---

## [Author Response · Author response to Decision Letter 1]

21 Apr 2025

Dear Editor

In name of all co-authors, it is my pleasure to submit our revised manuscript entitled “Quantitative assessment method for firefighting danger based on numerical simulation of forest fire spread in canyon wind fields” (PONE-D-24-40817).

Dear reviewers, thank you for reviewing our manuscript and providing constructive comments, which greatly assisted us in improving the manuscript. We have substantially revised our experiments. The manuscript has been carefully revised, and a point-by-point response to your comments is listed below. We hope that your concerns have been addressed accurately.

Taken together, we believe that the manuscript has been substantially improved and meets the standards required for publication in PLOS ONE. We look forward to your feedback.

Sincerely,

Dr. Wang

National Institute of Natural Hazards, Ministry of Emergency Management, Beijing, China

Response to Reviewer #1:

Reviewer’s General comments:

In the Methods section, the description of the input data is clearly insufficient. The conceptual risk assessment model does not make sense. Authors should start by conceptually defining the risk model based on the state of the art. This conceptual framework is critical. In order to be able to transform a generic "fire risk" model into a "firefighting risk" model, you must define the indicator classes based on suppression capacity. There are many inconsistencies and even errors in the use of fire science terminology. There are factors that enter the model twice. The way you defined the classes is wrong, it is disconnected from the physical process of fire propagation. It is also unclear how the factor weights were established. You simulated moments of a single fire (why this fire?). You did not describe the fire regime in your study area, nor the fire you used as a reference. You did not include the parameters used in the simulation, nor did you compare the simulated fire with the reference fire. This fire should have only been used to calibrate the parameters required by the simulator. The results should be evaluated using metrics such as overall accuracy, true skill statistics or others (see the work of Pontius on accuracy metrics). Then, these parameters would be used to define simulation scenarios. For each scenario, N fires would be simulated and the AHP would receive fire behavior metrics as input data, as well as the infrastructure frequency reached by the simulated fires.

You have a "Results and Discussion" section and a "Discussion" section, but the current discussion has not been done. To what extent does your proposal differ from others? Figures 8 and 9 are not understandable (why was the assessment not carried out for the entire study area?), and once again you are not consistent in terminology (what is "firewire strength"?). In the end, you have little more than a simulation that it is not even possible to know if it was done well.

The manuscript is very immature, and is not yet of sufficient quality to be considered for publication.

Reply: We sincerely thank the reviewers for their meticulous evaluation and invaluable feedback, which has guided substantial improvements to our manuscript. In response to your comments, we have implemented comprehensive revisions as follows:

In the Methods section, we provided detailed descriptions of input data to ensure spatial consistency across datasets. The conceptual framework for firefighting risk assessment was restructured with a focus on "firefighter safety risk," incorporating four core indicators—fire behavior, fuel complexity, terrain exposure, and operational exposure—to enhance model design. Terminology inconsistencies were systematically addressed (e.g., replacing "firewire strength" with the more precise term "fireline intensity").

For the fire process analysis, we added an in-depth explanation of the "3.28" Muli fire, clarifying its implications for risk assessment. Figures 8 and 9 now explicitly demonstrate that the actual burned area has undergone a risk assessment, while the unburned area remains untreated.

In the Discussion section, we expanded on the optimization of the wind field model and explored the accuracy of the proposed model. Additionally, we discussed current forest fire suppression methods and their implications for future research.

We deeply appreciate the reviewers' insights, which have significantly enhanced this work and inspired further exploration. We remain open to additional guidance and hope the revised manuscript meets the journal's rigorous standards.

Detailed comments:

Comment 1: L108: What do you mean by“climate is complex”?

Reply: Thank you for pointing out this ambiguity. We have revised the sentence to more precisely specify the climatic characteristics of the study area. The original phrase “climate is complex” has been replaced with: “In the research area, the climate is characterized by pronounced seasonal fluctuations and a high frequency of extreme weather phenomena” (Lines 163-164). This revision clarifies the climatic complexity by providing concrete descriptors, such as seasonal variability and the occurrence of extreme weather events, which are directly linked to fire risk.

Comment 2: L109: By "fire protection period" do you mean "fire season"? If so, what period covers the "fire season" in your geographic context?

Reply: We appreciate the suggestion to standardize terminology. The term “fire protection period” has been replaced with the more widely recognized term “fire season,” and the specific months are now explicitly stated: “especially marked by ongoing drought conditions during the fire season, which typically spans from January to May” (Lines 164-165). This adjustment aligns with regional fire management practices in southwestern Sichuan and enhances the clarity and consistency of the manuscript.

Comment 3: L110: Please change“The vegetation types primarily comprise (…)” by “The landscape is mainly covered by (…)”

Reply: Revised as suggested:“The landscape is mainly covered by evergreen coniferous forests...” (Lines 165)

Comment 4: L111: What do you meany by“serious swamping”? Please clarify. Will there be more peat bogs than swamps?

Reply: We apologize for the unclear terminology. The phrase “serious swamping” has been revised to specify the presence of peat bogs:“In addition, interforest meadows featuring remarkable peat bog development are prevalent, increasing the likelihood of underground fires” (Lines 167-168)

Comment 5:L112: Please change“underground fire occurrence” by“underground fires”

Reply: Revised as suggested: “increasing the likelihood of underground fires” (Lines 168)

Comment 6: L115: Change to“Input data and geoprocessing”

Reply: The section title has been updated to: “Input data and geoprocessing” (Lines 172)

Comment 7: L116-120: The description of the input data has to be more detailed. For example, it is important to mention what input data allowed generating the DEM, what methods were used to generate the derivatives that give rise to the slope and aspect (Horn's method?) and what software was used. The designation "land type" is unspecific, apparently it is a land cover map whose metadata must be minimally described (number of classes, method used, input data and auxiliary data that supported the construction of the dataset). Are meteorological data reanalysis? What scale are they on?

Reply: We have expanded the data description to include specific sources�processing methods and rewrote it: “Three fundamental data types were required to implement the forest fire spread and WindNinja models: digital elevation model (DEM) data, land cover data, and meteorological data. The DEM data were sourced from the Geographic Spatial Data Cloud (http://www.gscloud.cn) and then processed using spatial analysis techniques to yield slope and aspect data with a 30-meter spatial resolution. The land cover data were obtained from the global 30-meter spatial resolution fine land cover product (GLC_FCS30-2015) published by Liu Liangyun’s research team in 2019[25]. The meteorological data were collected from the European Centre for Medium-Range Weather Forecasts, using the recorded meteorological data from the nearest weather station to the Muli fire site (station number 56462; 29°N, 101.5°E) to inform the surface average wind field. This study also incorporated satellite imagery from Landsat 8 (http://www.gscloud.cn/) to accurately delineate areas affected by Muli fire. Before integrating this data into the model, it underwent a series of preprocessing steps, including radiometric correction, image registration, image cropping, and image mosaicking, to ensure uniform spatial extent across all datasets.”(Lines 173-185)

Comment 8:. L122: Change the title to “Wildire simulations”

Reply: The revised section title is now: “Wildfire simulations”

Comment 9:. L123: Add a reference of Mark Finney to FARSITE

Reply: A citation to Finney (1998) has been added: “FARSITE(Finney, 1998) is a two-dimensional fire modeling system that serves as a valuable supplementary decision-making instrument for the prevention and management of forest fires.” (Lines 189-190)

Comment 10:. L124: Add "fire" before "spread"

Reply: Revised as suggested: I removed “The Rothermel model and Huygens principle are two crucial theoretical bases of FARSITE that can simulate the spread of various types of fire under complex environmental conditions and obtain the simulation and prediction results of dozens of behavioral characteristic parameters during fire development.” as this was repetitive. This was described in the introduction.

Comment 11:. L124-126: Please rewrite to improve readability

Reply: Revised as suggested: I removed “The Rothermel model and Huygens principle are two crucial theoretical bases of FARSITE that can simulate the spread of various types of fire under complex environmental conditions and obtain the simulation and prediction results of dozens of behavioral characteristic parameters during fire development.” as this was repetitive. This was described in the introduction.

Comment 12:. L123: Add a reference to WindNinja

Reply: A citation to (Forthofer et al., 2014) has been added: “WindNinja (Forthofer et al., 2014) was used to produce high-resolution wind fields by incorporating the DEMs and meteorological data” (Lines 191-192)

Comment 13:. L127-129:"In addition" why? Simulations can be carried out with surface wind simulation or not. This choice is part of the simulation process, not exactly something independent.

Reply: We apologize for the ambiguous phrasing. The revised text clarifies the rationale: “WindNinja (Forthofer et al., 2014) was used to produce high-resolution wind fields by incorporating the DEMs and meteorological data. A significant benefit of WindNinja is its ability to integrate seamlessly with FARSITE, allowing for the direct application of the simulated wind fields.” (Lines 191-193)

Comment 14:. L129-130: Flammap already does this integration without having to import WindNinja outputs

Reply: We selected WindNinja for its superior performance in complex terrain. WindNinja generates spatially varying wind fields compatible with FARSITE.

Comment 15:. L131-138: Move to the previous section and rewrite to improve readability

Reply: The description of landscape file preparation has been moved to the “Comparison and selection of the forest fire behavior index” section and revised for clarity .

Comment 16:. L132: Change“slope”by“slope angle”and“slope direction” by“slope aspect”

Reply: Revised to standard terminology: “elevation, slope angle, and slope aspect”

Comment 17:. L135: What are the "classic fuel model values"? NFFL? Scott & Burgan? The reclassification of land cover to fuel models needs to be better described and should be supported by photographs.

Reply: Fuel models are now explicitly defined using the Scott and Burgan (2005) system: “The assignment of fuel types was conducted based on the 40-fuel model system established by Scott and Burgan (2005), including specific models such as M186 for high-load broadleaf litter.”(Lines 200-201)

Comment 18:. L142: Change“Forest fire behavior encompasses the various changes that occur throughout the fire-spreading process of forest fuel (…)”by“Wildfire behavior encompasses the spatial and temporal dynamics that occur throughout the fire-spreading process across different fuel complexes (…)”

Reply: Revised as suggested: “Wildfire behaviors encompass the spatial and temporal dynamics that occur during the fire-spreading process across different fuel complexes...” (Lines 205-206)

Comment 19:. L143-144: Please be careful in your use of terminology:

"spreading speed of the fire head" = "fire rate of spread"

"fire intensity" = "fireline intensity"

"firefield scope expansion" = ?

"other extreme fire behaviors" = ? (probability of crown fire?)

Reply: Revised to standard terminology: “Wildfire behavior is characterized by several factors, including the fire rate of spread, the expansion of the fire perimeter, the escalation of the fireline intensity, and extreme fire behaviors such as spotting, fire whirls, and flare-ups” (Lines 207-209)

Comment 20:. L147: The flame length is directly proportional to the fireline intensity, but not to the fire rate of spread.

Reply: We thank the reviewer for this correction. The sentence has been revised to reflect this relationship:“The spread rate of a wildfire, recognized as a fundamental indicator of fire behaviors, significantly impacts the development and expansion of the fire. Furthermore, it is an essential metric for evaluating the dynamic alterations that occur within the fire environment during firefighting efforts. This spread rate also serves as a primary criterion for fire command personnel when organizing firefighting teams.” (Lines 211-215)

Comment 21:. L154-156: Move to the next section

Reply: The text describing the AHP method has been moved to the “Firefighting risk assessment model” section

Comment 22:. L158-159: You state that“fire behavior, forest fuel factor, topography, and exposure were identified as the four first-level indexes”. How? What other factors did you consider? Topography and fuels are already integrated into fire behavior indicators. You are "doubling" the weight of these factors in the analytical process.

Reply: We sincerely appreciate the reviewer’s insightful critique regarding the selection of first-level indices. We acknowledge that fire behavior inherently incorporates interactions between fuels and topography. However, in designing this firefighting risk assessment model, our goal was to explicitly isolate the unique contributions of each factor to operational safety, rather than relying solely on composite fire behavior metrics. While fuels and topography influence fire behavior, their operational impacts extend beyond combustion physics. For example, a high-slope area with moderate fire intensity may still pose extreme risks due to limited escape routes—a dimension not captured by fire behavior indices alone. By separating these factors, we ensure that suppression capacity and safety are explicitly weighted in the risk calculus.

Comment 23:. L160: You have to maintain consistency in the terminology used

Reply: We apologize for inconsistencies. Terminology has been standardized (e.g., “fuel type” → “fuel model”, “slope” → “slope angle”) throughout the manuscript.

Comment 24:. L161: What do you mean by“secondary evaluation indexes”?

Reply: The term “secondary evaluation indexes” refers to sub-indicators under each first-level index. For clarity, we now use “secondary indicators”.

Comment 25:. L161:“Fuel type”or fuel models?

Reply: Revised to “fuel model” to align with the Scott and Burgan (2005) .

Comment 26:. L161:“Fuel type and intensity”? What’s“fuel intensity”?

Reply: We apologize for the ambiguity. “Fuel intensity” has been replaced with “fuel density”, defined as “Number of plants per unit area”.

Comment 27:. L162: "Slope" and "slope inclination" are basically the same. Maybe you mean "slope angle" and "slope aspect"

Reply: Revised to standard terms: “slope angle and slope aspect”.

Comment 28:. L163-164: The distance to a burnt edge is not exactly a measure of exposure. Infrastructure exposure can be assessed by the fre

---

## [Decision Letter · Decision Letter 1]

Dear Dr. Wang,

**Please address the comments raised by Reviewer 3.**

We look forward to receiving your revised manuscript.

Kind regards,

Isidoro Russo, Ph.D.

Academic Editor

PLOS ONE

**Journal Requirements:**

Reviewers' comments:

Reviewer's Responses to Questions

**Comments to the Author**

Reviewer #1: (No Response)

Reviewer #2: All comments have been addressed

Reviewer #3: All comments have been addressed

2. Is the manuscript technically sound, and do the data support the conclusions?

Reviewer #1: No

Reviewer #2: Yes

Reviewer #3: Yes

3. Has the statistical analysis been performed appropriately and rigorously?

Reviewer #1: No

Reviewer #2: Yes

Reviewer #3: Yes

4. Have the authors made all data underlying the findings in their manuscript fully available?

Reviewer #1: No

Reviewer #2: Yes

Reviewer #3: Yes

5. Is the manuscript presented in an intelligible fashion and written in standard English?

Reviewer #1: No

Reviewer #2: Yes

Reviewer #3: Yes

**Reviewer #1:**  Manuscript title: Quantitative assessment method for firefighting danger based on numerical simulation of forest fire spread in canyon wind fields

Manuscript reference: PONE-D-24-40817R1

General comments:

The integration of WindNinja and FARSITE is useful and technically sound, but it's not novel by itself. Several studies - some even cited in your manuscript (e.g., Sibanda et al., Forthofer et al. ) - have already employed this coupling for simulating fire behavior in complex terrain. To strengthen the originality claim, you should emphasize how your application differs - e.g.: [1] specific regional application (Muli Canyon terrain, China); [2] a customized risk assessment model using AHP; [3] quantitative validation of canyon wind field simulations using field observations. If these are your contributions, they need to be clearly positioned in the Introduction and Discussion as “novel aspects” that build upon (not replicate) existing work.

In the Introduction section, the poor choice of references is evident, and I make some suggestions in the Specific comments. The Introduction could benefit from sharper focus. Currently, it blends general background with case-specific information. You need to clearly separate general wildfire context, the knowledge gap, and the specific aims of your study.

In the Materials and Methods section, the authors did not address many of the issues I raised in my previous review. There is a notable gap in the description of fire-weather inputs and fuel moisture, which are critical to FARSITE simulations:

Live and dead fuel moisture content: not reported, yet these are critical input parameters.

WindNinja configuration: limited description of input stability class and simulation mode (e.g., diurnal mode, stable mode).

Fuel model customization: While the manuscript mentions using Scott and Burgan’s 40 fuel models, it lacks detail on how well those models match local Chinese vegetation or if any modifications were applied.

In Table 2 of your manuscript, the Analytic Hierarchy Process (AHP) assigns independent weights to fire behavior metrics (spread rate, fireline intensity, flame length); fuel factors (fuel type, fuel density – this last one is a proxy of fuel load); and terrain factors (slope angle, slope aspect); but fire behavior metrics are not independent of fuel and terrain, and thus, you’re double-counting such effects. Furthermore, the authors are unaware of the relationships between fire behavior indicators. Flame length and fireline intensity are highly correlated. Here they have some empirical formulas:

L=0.0775I_B^0.46 (Byram, 1959)

L=0.0276I_B^(2⁄3) (Thomas, 1963)

I_B=246.0L^1.711 (Rossa et al., 2024) – Forests and shrublands

I_B=574.2L^1.956 (Rossa et al., 2024) – Grasslands

I_B=300L^2 Generic rule

Just use one of the metrics, otherwise you are overestimating the relative weight of metrics related to energy release.

In a AHP process this is a problem due to theoretical redundancy. In a properly constructed AHP, input variables should be independent, and if A causes B, including both A and B violates AHP’s assumptions. Weighting bias is also a problem, since you're giving disproportionate influence to some metrics and inflating danger index in areas where those factors dominate, regardless of actual modeled fire behavior.

The conceptual risk assessment model does not make sense and to solve the redundancy between fire behavior outputs and their underlying drivers (fuels, terrain), the model should be split into two tiers: a) Fire Behavior Simulation (Output-Based Risk) including Rate of Spread and Fireline Intensity (composite Fire Behavior Danger Index - FBDI); b) Exposure & Vulnerability Assessment (AHP-Based) assessing how challenging it would be to suppress the fire, independent of fire characteristics through exposure (distance to settlements and distance to water sources) and suppression difficulty (road accessibility, barriers to operational resources) (Operational Suppression Danger Index - OSDI). The final value can be obtained through a formula such as: Total Firefighting Danger Index (TFDI) = α × FBDI + β × OSDI where α, β are determined via expert consultation or empirical validation. You should still be aware that class definition should be aware of the effects of fire or suppression capacity.

Your work still has major limitations due to the fact that you only used one fire to determine all of these effects. A model built with so little information is not very robust. Since this fire is not contextualized in a fire regime, it is not even possible to know whether it is representative of the local reality. On the other hand, you only compare the wind characteristics between the simulation with WindNinja and what was measured at the stations, demonstrating that you achieve better accuracy in the perimeter of the various stages of progression of this fire. But how were these stages determined? The satellite images you refer to do not have the temporal resolution to allow this daily definition. What other data did you use?

The Discussion section is still very poor and needs to be strengthened.

You need to check language and grammas, since numerous grammatical issues are present, including awkward phrasing, redundancy, occasional tense shifts, and overuse of passive voice. A full professional English language editing pass is essential.

Specific comments:

L41-43: The reference used does not support your claim. It is just a case study in India. If you want to highlight changes in fire regimes, highlight the increased frequency of very destructive wildfires, or fires with extreme behavior, it seems to me more appropriate to use some of the references I suggest below:

Canada: https://doi.org/10.1139/cjfr-2023-0298;
https://doi.org/10.1139/cjfr-2024-0092;
https://doi.org/10.1038/s43247-023-00977-1

United States of America: https://doi.org/10.1111/geb.13496;
https://doi.org/10.1029/2021GL097131;
https://doi.org/10.1088/1748-9326/abae9e

Australia: https://doi.org/10.1088/1748-9326/abeb9e;
https://doi.org/10.1029/2020EF001884;
https://doi.org/10.3390/fire4040097;
https://doi.org/10.3390/fire6110438;
https://doi.org/10.1111/gcb.15125

Greece: https://doi.org/10.3390/su13031556;
https://doi.org/10.3390/fire7120467;

But also Portugal: https://doi.org/10.14195/978-989-26-16-506_48;
https://doi.org/10.1088/1748-9326/ac8be4;
https://doi.org/10.1016/j.isci.2023.106141;
https://doi.org/10.3390/fire3040057;
https://doi.org/10.5194/nhess-22-4019-2022;
https://doi.org/10.1007/s10021-016-0010-2;
https://doi.org/10.1002/2016JG003389; in Brazil: https://doi.org/10.1126/sciadv.aay1632; in Chile: https://doi.org/10.1016/j.wace.2024.100716; or even in Korea: https://doi.org/10.1016/j.agrformet.2024.109920

I also recommend reading this paper: https://doi.org/10.5194/essd-16-3601-2024

L42: Remove double “,”

L44: Change “unpredictable fire behaviors” to “unpredictable fire behavior”

L44-46: Rewrite the sentence, the unpredictability of fire behavior depends on the interaction between different factors. As it is written it seems independent of the others.

L46-47: Again, your reference is not the most appropriate. See the following works: https://doi.org/10.1071/WF17114;
https://doi.org/10.1071/WF17147;
https://doi.org/10.3390/fire2040052;
https://doi.org/10.1071/WF19022;
https://doi.org/10.3390/fire2030040;
http://dx.doi.org/10.1071/WF13021;
https://doi.org/10.1080/00049158.2001.10676160;
https://doi.org/10.1071/WF16213;
https://doi.org/10.1016/j.firesaf.2008.01.001

L48: Change “spread of forest fires” to “fire spread”

L50-51: But the cited paper only analyzed the distribution of fires in China between 2003 and 2016, and you refer in the sentence to events up to 2022.

L62: Change “forest fire” to “wildfire”

L62-65: Please, when you start a narrative about fire spread models, cite the seminal works. Rothermel and Finney have a prominent place in fire science. After that, I agree, it may make sense to highlight some examples of application. Note that McArthur's model is a fire-weather danger model like the Canadian Fire Weather Index, and not a fire spread model.

L65-68: Has anyone evaluated this effect? Is the uncertainty related to wind greater than that resulting from a misassignment of the fuel model?

L66-69: See the following papers: https://doi.org/10.1016/j.scitotenv.2016.06.112;
https://doi.org/10.1016/j.scitotenv.2017.03.106;
https://doi.org/10.1186/s40064-016-2842-9;
https://doi.org/10.3390/rs8040326

L74-75: Change “more intricate models that rely specifically on physical processes” to “physical-based models”, and at least provide references for those you give as examples. You should take some time to read the work of Andrew Sullivan and Miguel Cruz, both researchers at CSIRO.

L76-78: And?

L77: Change “forest fire behaviors” to “wildfire behavior”

L78-80: And?

L80: I did not find Seungmin et al. in the references.

L82-83: You mention that “These studies have contributed to the development of more effective fire response strategies (…)”. Where? Are you sure about that? At most they may be useful to improve the response, but do practitioners use these findings and tools in their decision-making process?

L95-98: Please clarify. Created a fire spread model? Based on what?

L98-99: What Reimer et al. did was evaluate the effectiveness of the initial attack.

L103: Change “fire behaviors” to “fire behavior” (and alsewhere in the manuscript)

L105: Change “canyon topography” to “canyons” (there will be no consensus on this type of topography)

L109-115: But will these models provide the appropriate response when it comes to fire safety? To me, they seem to have a scale and principles that are not suited to such a specific issue. See, for exemple https://doi.org/10.1071/WF13063

L116-121: I have given some examples above. There is indeed little research done in this area, but it is not limited to what you describe.

L122-126: I didn't understand both sentences.

L130: “firefighting dangers” sounds awkward.

L132: You didn't develop it, you tested it, which is quite different.

L135: “Fuel characteristics” and “land resources” are already included in “fire behavior”.

L150: Why remarkable? The fire consumed 230ha, which is not a large amount when compared to the large, extreme fires that typically spread in fire-prone regions. It spread over four days, so the fire spread and the rate of expansion were not high. There was a period of crown fire, but that is not what makes this fire particularly significant.

L150: Change “ignited” by “started”

L161-169: This paragraph should be before the event description.

L172-174: You did not properly answer my concerns in the previous review. You only say where you extracted the DEM from. You do not mention how the DEM was determined (satellite imagery? If so, from which sensor?), and you do not mention the software used in the processing, and what algorithms you used to derive the slope angle and slope aspect.

L180: You refer to baseline data for “correctly delineated areas affected by the Muli fire,” but not how you processed that baseline data. Did you use vegetation indices?

L186-191: You again fail to mention all the input parameters used in the simulation, in particular with regard to fuel moisture.

L192-218: The title of this subsection does not correspond to its content.

L198-200: You do not mention the specific correspondence between land cover classes and fuel models, and you do not justify the correspondence you made.

L212-218: Flame length and fireline intensity are highly correlated with each other.

L220: Change “dangers” to “fire danger” (and elsewhere in the manuscript)

L220-227: I continue to state that there is a problem of redundancy in the variables. Fire behavior metrics already integrate all the others. This formulation makes no sense.

L236-237: It is anything but comprehensible, both the scale (which is not Saaty) and Table 1 itself.

L256: How many experts were surveyed? What is their profile?

L266: Remove “gradually increasing”

L308-310: I believe that a fire of just over 200ha is not enough for the authors' conclusion. Even so, an improvement was expected. So far, nothing new.

L366: It is not clear in the methods of how you determined these indices in Table 4. Figure 1 is cryptic and the methodological description does not allow for replication.

**Reviewer #2: ** (No Response)

**Reviewer #3: ** This revised version is suitable for publication.

1. Only small comment, Keywords: It is crucial to revise the keywords, ensuring they are spelled correctly and avoid general, abbreviations, and plural terms and multiple concepts (avoid, for example, 'and', 'of'). This will help to maintain the precision and clarity of the manuscript.

e.g. assessment of firefighting danger (too long)

2. In the main text, many numeric data are given too many significant figures; two significant figures suffice, and three suffice if the first significant figure is "1."

3. You must provide all the figures in high resolution and make the labels and legends more legible.

4. Conclusion: The findings could be further developed; the article contains a lot of interesting data.

**Do you want your identity to be public for this peer review?** For information about this choice, including consent withdrawal, please see our Privacy Policy

Reviewer #1: **Yes: ** Nuno Ricardo Gracinhas Nunes Guiomar

Reviewer #2: No

Reviewer #3: No

---

## [Author Response · Author response to Decision Letter 2]

3 Jul 2025

Dear Editor

In name of all co-authors, it is my pleasure to submit our revised manuscript entitled “Quantitative assessment method for firefighting danger based on numerical simulation of forest fire spread in canyon wind fields” (PONE-D-24-40817R1).

Dear reviewers, thank you for reviewing our manuscript and providing constructive comments, which greatly assisted us in improving the manuscript. We have substantially revised our experiments. The manuscript has been carefully revised, and a point-by-point response to your comments is listed below. We hope that your concerns have been addressed accurately.

Taken together, we believe that the manuscript has been substantially improved and meets the standards required for publication in PLOS ONE. We look forward to your feedback.

Sincerely,

Dr. Wang

National Institute of Natural Hazards, Ministry of Emergency Management, Beijing, China

Response to Reviewer #3:

Detailed comments:

Comment 1: Only small comment, Keywords: It is crucial to revise the keywords, ensuring they are spelled correctly and avoid general, abbreviations, and plural terms and multiple concepts (avoid, for example, 'and', 'of'). This will help to maintain the precision and clarity of the manuscript.

Reply: Thank you for this valuable suggestion to enhance keyword precision. We have carefully refined the keywords as follows:

Forest firefighting danger; Canyon wind field; Fire spread simulation; FARSITE; WindNinja; Risk assessment

Comment 2: In the main text, many numeric data are given too many significant figures; two significant figures suffice, and three suffice if the first significant figure is "1."

Reply: We sincerely appreciate this meticulous observation. We have now comprehensively adjusted numerical values throughout the manuscript:

Table 2. Weights of forest firefighting danger assessment index.

Primary index Primary index weight

Forest fire behavior 0.28

Fuel factor 0.48

Terrain factor 0.17

Exposure 0.06

Secondary index Secondary index weight

Spread rate 0.61

Fireline intensity 0.30

Flame length 0.09

Fuel type 0.83

Fuel density 0.17

Slope angle 0.83

Slope aspect 0.17

Distance to town 0.25

Distance to the water source 0.75

Table 3. Comparison of simulation accuracy of different wind fields.

Wind field Time Area of coincidence area /km2 Over-simulated area /km2 Unsimulated area /km2 SC coefficient

Average surface wind field March 29, 19:00 60 19 6.1 0.82

March 30, 19:00 86 59 10.8 0.71

April 1, 19:00 78 204 22 0.40

WindNinja wind field March 29, 19:00 62 4.4 3.9 0.94

March 30, 19:00 86 36 12 0.78

April 1, 19:00 80 178 21 0.44

Comment 3: You must provide all the figures in high resolution and make the labels and legends more legible.

Reply: We gratefully acknowledge this important recommendation. To improve visual clarity:

All figures regenerated at 600 dpi resolution.

Enhanced color contrast in categorical displays.

Supplementary png files provided.

Specific improvements:

Fig 1. Assessment model of forest firefighting danger level.

Fig 2. Wind field data from March 25 to April 3, 2020 in the research area.

Fig 3. Comparison between simulated wind field and actual wind field.

Comment 4: Conclusion: The findings could be further developed; the article contains a lot of interesting data.

Reply: We deeply appreciate your encouraging feedback on our data. The conclusion has been substantially enriched:

To address the limitations of conventional forest fire spread models for predicting fire behavior in complex terrains such as mountainous and valley regions, this study introduces a coupled simulation approach that integrates the FARSITE forest fire spread simulation system with the high-resolution wind field model WindNinja. Furthermore, a forest firefighting danger assessment model was developed, incorporating factors such as forest fire behavior, fuel characteristics, topographical features, and exposure indices. This model was used to quantify the firefighting danger associated with the Muli forest fire that occurred on March 28 in Liangshan, Sichuan Province. The key findings, significantly enriched by the detailed simulation data, offer a robust scientific foundation for implementing targeted forest firefighting strategies and address the core research question on enhancing danger quantification in canyon terrains:

(1)The wind field within the canyon displays a significant diurnal variation, with maximum wind speeds generally recorded between 2 PM and 4 PM. A comparative assessment of the simulated wind speeds against actual local meteorological conditions revealed inconsistencies, with an average deviation of approximately 2 m/s in the simulated data. However, the wind direction exhibits a degree of stability, predominantly blowing from the southwest (SW) and west-southwest (WSW). The high-resolution wind field data produced by the WindNinja model demonstrates its efficacy in accurately simulating the dynamics of forest fire propagation in relation to the canyon's topographical characteristics.

(2) The simulation indicated concerning levels of fire behavior risk, with regions categorized as having high and extremely dangerous fire spread rates accounting for 16% and 54% of the study area, respectively, primarily located in the central and southern regions. Although areas exhibiting high-danger fire line intensity were notable at 5.9%, extreme-danger intensity zones, while less prevalent at 0.14%, pose significant flashpoint hazards. Additionally, high and extreme-danger flame lengths encompassed 21% and 8.1% of the area, respectively, demonstrating a strong spatial correlation with high spread rates. This quantitative spatial mapping of zones exhibiting extreme fire behavior represents a significant finding of the integrated model.

(3) The analysis of fuel types has identified grasslands, deciduous broadleaf forests, and evergreen coniferous forests as the primary combustible materials. Notably, grasslands demonstrated the highest average fire behavior index (11) and an overall firefighting danger index (10.1), followed by deciduous broadleaf forests with indices of 10.3 and 8.4, respectively. These values significantly surpass the danger levels associated with evergreen coniferous forests, which recorded indices of 7.4 and 6.0. This finding highlights the considerable risk posed by grassland fuels in the region. Additionally, exposure analysis revealed that the threat of firefighting danger escalates with increasing distance from urban areas or water sources.

(4) Upon synthesizing all relevant factors, the thorough danger assessment indicated that a significant 37% of the Muli fire site area is subjected to high or extremely high levels of firefighting danger, with concentrations predominantly located in the central and western regions. An analysis based on forest fuel types clearly identified grassland ecosystems as the vegetation type with the highest associated risk.

At the Muli fire site, 38% of the area is classified as having a high or extremely high danger of fire, with this danger predominantly located in the central and western regions of the site. An examination of forest fuel classifications reveals that grassland ecosystems represent the highest-risk vegetation type, exhibiting a comprehensive danger index of 10.1. Areas classified as extreme danger are predominantly located in central and western canyons, where southwesterly winds exceeding 18 m/s have been shown to increase fire spread rates by 54%. To reduce the risk of firefighter casualties, we propose the following recommendations: (1) The real-time integration of WindNinja-FARSITE forecasts into firefighting operations. (2) The implementation of preemptive fuel reduction strategies in grasslands situated within a 5 km radius of urban areas. (3) The tactical avoidance of leeward slopes during peak diurnal wind periods, specifically between 14:00 and 16:00.

Response to Editor:

Thank you for raising this important consideration regarding image copyright. We would like to clarify that all cartographic representations in our study were generated through original processing of open-access Landsat base data (Landsat: http://landsat visibleearth.nasa.gov/). At the same time, we have made the following adjustments to the picture

Fig 4. WindNinja wind field and fire line intensity coupling simulation.

Fig 5. Simulation results of fire behavior and exposure in the research area.

Fig 6. Forest fire comprehensive danger level map.

We are deeply grateful for your insightful suggestions, which have significantly strengthened our work. All modifications have been meticulously implemented, with tracked changes provided for your convenience. Should any further clarifications be needed, we remain available to address them promptly.

With best wishes,

Sincerely yours,

Ao Wang , Chenghu Wang , Guiyun Gao , Ningyu Wu , Haiyan Su

---

## [Editor Report · Decision Letter 2]

Quantitative assessment method for firefighting danger based on numerical simulation of forest fire spread in canyon wind fields

PONE-D-24-40817R2

Dear Dr. Chenghu Wang,

We’re pleased to inform you that your manuscript has been judged scientifically suitable for publication and will be formally accepted for publication once it meets all outstanding technical requirements.

Kind regards,

Isidoro Russo, Ph.D.

Academic Editor

PLOS ONE
---

## [Editor Report · Acceptance letter]

PONE-D-24-40817R2

PLOS ONE

Dear Dr. Wang,

I'm pleased to inform you that your manuscript has been deemed suitable for publication in PLOS ONE. Congratulations! Your manuscript is now being handed over to our production team.

Kind regards,

on behalf of

Dr. Isidoro Russo

Academic Editor

PLOS ONE